# AUGMIX: A SIMPLE DATA PROCESSING METHOD TO IMPROVE ROBUSTNESS AND UNCERTAINTY

**Dan Hendrycks**[*]
DeepMind
hendrycks@berkeley.edu

**Norman Mu**[*]
Google
normanmu@google.com

**Ekin D. Cubuk**
Google
cubuk@google.com

**Barret Zoph**
Google
barretzoph@google.com

**Justin Gilmer**
Google
gilmer@google.com

**Balaji Lakshminarayanan**[†]
DeepMind
balajiln@google.com

## ABSTRACT

Modern deep neural networks can achieve high accuracy when the training distribution and test distribution are identically distributed, but this assumption is frequently violated in practice. When the train and test distributions are mismatched, accuracy can plummet. Currently there are few techniques that improve robustness to unforeseen data shifts encountered during deployment. In this work, we propose a technique to improve the robustness and uncertainty estimates of image classifiers. We propose AUGMIX, a data processing technique that is simple to implement, adds limited computational overhead, and helps models withstand unforeseen corruptions. AUGMIX significantly improves robustness and uncertainty measures on challenging image classification benchmarks, closing the gap between previous methods and the best possible performance in some cases by more than half.

## 1 INTRODUCTION

Current machine learning models depend on the ability of training data to faithfully represent the data encountered during deployment. In practice, data distributions evolve (Lipton et al., 2018), models encounter new scenarios (Hendrycks & Gimpel, 2017), and data curation procedures may capture only a narrow slice of the underlying data distribution (Torralba & Efros, 2011). Mismatches between the train and test data are commonplace, yet the study of this problem is not. As it stands, models do not robustly generalize across shifts in the data distribution. If models could identify when they are likely to be mistaken, or estimate uncertainty accurately, then the impact of such fragility might be ameliorated. Unfortunately, modern models already produce overconfident predictions when the training examples are independent and identically distributed to the test distribution. This overconfidence and miscalibration is greatly exacerbated by mismatched training and testing distributions.

Small corruptions to the data distribution are enough to subvert existing classifiers, and techniques to improve corruption robustness remain few in number. Hendrycks & Dietterich (2019) show that classification error of modern models rises from 22% on the usual ImageNet test set to 64% on ImageNet-C, a test set consisting of various corruptions applied to ImageNet test images. Even methods which aim to explicitly quantify uncertainty, such as probabilistic and Bayesian neural networks, struggle under data shift, as recently demonstrated by Ovadia et al. (2019). Improving performance in this setting has been difficult. One reason is that training against corruptions only encourages networks to memorize the specific corruptions seen during training and leaves models unable to generalize to new corruptions (Vasiljevic et al., 2016; Geirhos et al., 2018). Further, networks trained on translation augmentations remain highly sensitive to images shifted by a single pixel (Gu et al., 2019; Hendrycks & Dietterich, 2019). Others have proposed aggressive data augmentation schemes (Cubuk et al., 2018), though at the cost of a computational increase. Chun et al. (2019) demonstrates that many techniques may improve clean accuracy at the cost of robustness while many techniques which improve robustness harm uncertainty, and contrariwise. In all, existing techniques have considerable trade-offs.

---

[*]Equal Contribution.
[†]Corresponding author.

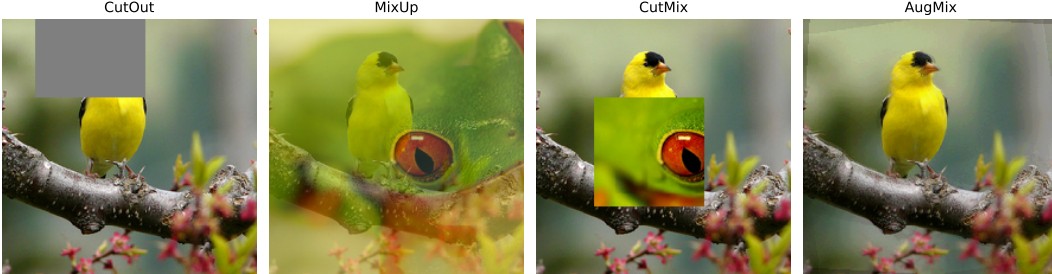

Figure 1: A visual comparison of data augmentation techniques. AUGMIX produces images with variety while preserving much of the image semantics and local statistics.

In this work, we propose a technique to improve both the robustness and uncertainty estimates of classifiers under data shift. We propose AUGMIX, a method which simultaneously achieves new state-of-the-art results for robustness and uncertainty estimation while maintaining or improving accuracy on standard benchmark datasets. AUGMIX utilizes stochasticity and diverse augmentations, a Jensen-Shannon Divergence consistency loss, and a formulation to mix multiple augmented images to achieve state-of-the-art performance. On CIFAR-10 and CIFAR-100, our method roughly halves the corruption robustness error of standard training procedures from 28.4% to 12.4% and 54.3% to 37.8% error, respectively. On ImageNet, AUGMIX also achieves state-of-the-art corruption robustness and decreases perturbation instability from 57.2% to 37.4%. Code is available at https://github.com/google-research/augmix.

## 2 RELATED WORK

**Robustness under Data Shift.** Geirhos et al. (2018) show that training against distortions can often fail to generalize to unseen distortions, as networks have a tendency to memorize properties of the specific training distortion. Vasiljevic et al. (2016) show training with various blur augmentations can fail to generalize to unseen blurs or blurs with different parameter settings. Hendrycks & Dietterich (2019) propose measuring generalization to unseen corruptions and provide benchmarks for doing so. Kang et al. (2019) construct an adversarial version of the aforementioned benchmark. Gilmer et al. (2018); Gilmer & Hendrycks (2019) argue that robustness to data shift is a pressing problem which greatly affects the reliability of real-world machine learning systems.

**Calibration under Data Shift.** Guo et al. (2017); Nguyen & O'Connor (2015) propose metrics for determining the calibration of machine learning models. Lakshminarayanan et al. (2017) find that simply ensembling classifier predictions improves prediction calibration. Hendrycks et al. (2019a) show that pre-training can also improve calibration. Ovadia et al. (2019) demonstrate that model calibration substantially deteriorates under data shift.

**Data Augmentation.** Data augmentation can greatly improve generalization performance. For image data, random left-right flipping and cropping are commonly used He et al. (2015). Random occlusion techniques such as Cutout can also improve accuracy on clean data (Devries & Taylor, 2017; Zhong et al., 2017). Rather than occluding a portion of an image, CutMix replaces a portion of an image with a portion of a different image (Yun et al., 2019). Mixup also uses information from two images. Rather than implanting one portion of an image inside another, Mixup produces an elementwise convex combination of two images (Zhang et al., 2017; Tokozume et al.,

Figure 2: Example ImageNet-C corruptions. These corruptions are encountered only at test time and not during training.

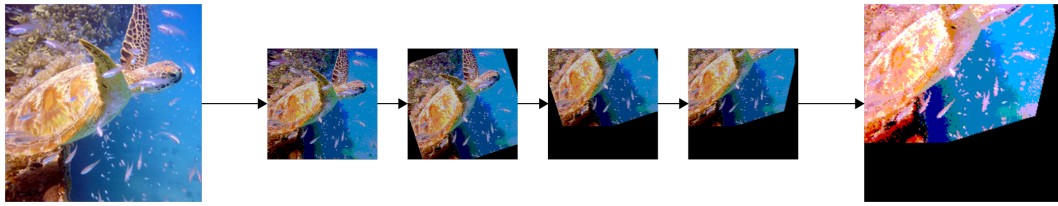

Figure 3: A cascade of successive compositions can produce images which drift far from the original image, and lead to unrealistic images. However, this divergence can be balanced by controlling the number of steps. To increase variety, we generate multiple augmented images and mix them.

2018). Guo et al. (2019) show that Mixup can be improved with an adaptive mixing policy, so as to prevent manifold intrusion. Separate from these approaches are learned augmentation methods such as AutoAugment (Cubuk et al., 2018), where a group of augmentations is tuned to optimize performance on a downstream task. Patch Gaussian augments data with Gaussian noise applied to a randomly chosen portion of an image (Lopes et al., 2019). A popular way to make networks robust to $\ell_p$ adversarial examples is with adversarial training (Madry et al., 2018), which we use in this paper. However, this tends to increase training time by an order of magnitude and substantially degrades accuracy on non-adversarial images (Raghunathan et al., 2019).

## 3 AUGMIX

AUGMIX is a data augmentation technique which improves model robustness and uncertainty estimates, and slots in easily to existing training pipelines. At a high level, AugMix is characterized by its utilization of simple augmentation operations in concert with a consistency loss. These augmentation operations are sampled stochastically and layered to produce a high diversity of augmented images. We then enforce a consistent embedding by the classifier across diverse augmentations of the same input image through the use of Jensen-Shannon divergence as a consistency loss.

Mixing augmentations allows us to generate diverse transformations, which are important for inducing robustness, as a common failure mode of deep models in the arena of corruption robustness is the memorization of fixed augmentations (Vasiljevic et al., 2016; Geirhos et al., 2018). Previous methods have attempted to increase diversity by directly composing augmentation primitives in a chain, but this can cause the image to quickly degrade and drift off the data manifold, as depicted in Figure 3. Such image degradation can be mitigated and the augmentation diversity can be maintained by mixing together the results of several augmentation chains in convex combinations. A concrete account of the algorithm is given in the pseudocode below.

---

**Algorithm** AUGMIX Pseudocode

1: **Input:** Model $\hat{p}$, Classification Loss $\mathcal{L}$, Image $x_{\text{orig}}$, Operations $\mathcal{O} = \{\text{rotate}, \dots, \text{posterize}\}$
2: **function** AugmentAndMix($x_{\text{orig}}$, $k = 3$, $\alpha = 1$)
3:     Fill $x_{\text{aug}}$ with zeros
4:     Sample mixing weights $(w_1, w_2, \dots, w_k) \sim \text{Dirichlet}(\alpha, \alpha, \dots, \alpha)$
5:     **for** $i = 1, \dots, k$ **do**
6:         Sample operations $\text{op}_1, \text{op}_2, \text{op}_3 \sim \mathcal{O}$
7:         Compose operations with varying depth $\text{op}_{12} = \text{op}_2 \circ \text{op}_1$ and $\text{op}_{123} = \text{op}_3 \circ \text{op}_2 \circ \text{op}_1$
8:         Sample uniformly from one of these operations chain $\sim \{\text{op}_1, \text{op}_{12}, \text{op}_{123}\}$
9:         $x_{\text{aug}} \mathrel{+}= w_i \cdot \text{chain}(x_{\text{orig}})$               ▷ *Addition is elementwise*
10:    **end for**
11:    Sample weight $m \sim \text{Beta}(\alpha, \alpha)$
12:    Interpolate with rule $x_{\text{augmix}} = m x_{\text{orig}} + (1 - m) x_{\text{aug}}$
13:    **return** $x_{\text{augmix}}$
14: **end function**
15: $x_{\text{augmix1}} = \text{AugmentAndMix}(x_{\text{orig}})$         ▷ *$x_{augmix1}$ is stochastically generated*
16: $x_{\text{augmix2}} = \text{AugmentAndMix}(x_{\text{orig}})$              ▷ *$x_{augmix1} \neq x_{augmix2}$*
17: **Loss Output:** $\mathcal{L}(\hat{p}(y \mid x_{\text{orig}}), y) + \lambda \,\text{Jensen-Shannon}(\hat{p}(y \mid x_{\text{orig}}); \hat{p}(y | x_{\text{augmix1}}); \hat{p}(y | x_{\text{augmix2}}))$

---

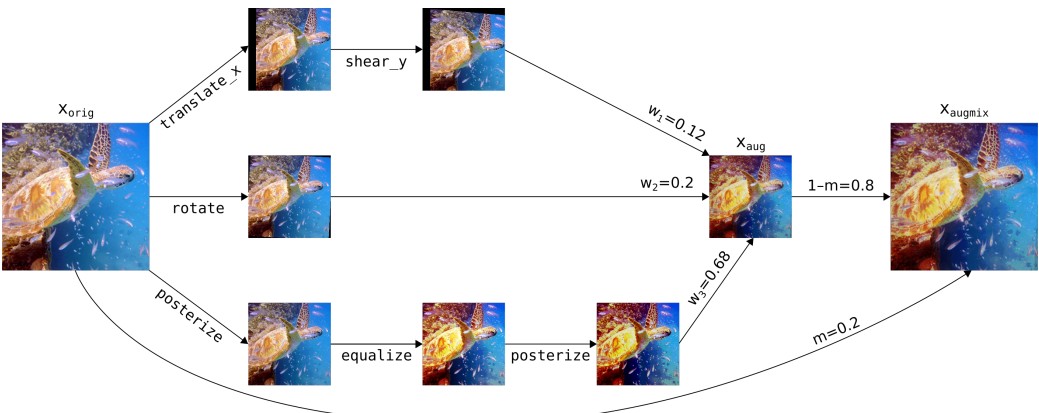

Figure 4: A realization of AUGMIX. Augmentation operations such as translate_x and weights such as $m$ are randomly sampled. Randomly sampled operations and their compositions allow us to explore the semantically equivalent input space around an image. Mixing these images together produces a new image without veering too far from the original.

**Augmentations.** Our method consists of mixing the results from augmentation chains or compositions of augmentation operations. We use operations from AutoAugment. Each operation is visualized in Appendix C. Crucially, *we exclude operations which overlap with ImageNet-C corruptions*. In particular, we remove the `contrast`, `color`, `brightness`, `sharpness`, and `Cutout` operations so that our set of operations and the ImageNet-C corruptions are disjoint. In turn, we do not use any image noising nor image blurring operations so that ImageNet-C corruptions are encountered only at test time. Operations such as rotate can be realized with varying severities, like $2°$ or $-15°$. For operations with varying severities, we uniformly sample the severity upon each application. Next, we randomly sample $k$ augmentation chains, where $k = 3$ by default. Each augmentation chain is constructed by composing from one to three randomly selected augmentation operations.

**Mixing.** The resulting images from these augmentation chains are combined by mixing. While we considered mixing by alpha compositing, we chose to use elementwise convex combinations for simplicity. The $k$-dimensional vector of convex coefficients is randomly sampled from a Dirichlet$(\alpha, \ldots, \alpha)$ distribution. Once these images are mixed, we use a "skip connection" to combine the result of the augmentation chain and the original image through a second random convex combination sampled from a Beta$(\alpha, \alpha)$ distribution. The final image incorporates several sources of randomness from the choice of operations, the severity of these operations, the lengths of the augmentation chains, and the mixing weights.

**Jensen-Shannon Divergence Consistency Loss.** We couple with this augmentation scheme a loss that enforces smoother neural network responses. Since the semantic content of an image is approximately preserved with AUGMIX, we should like the model to embed $x_{\text{orig}}$, $x_{\text{augmix1}}$, $x_{\text{augmix2}}$ similarly. Toward this end, we minimize the Jensen-Shannon divergence among the posterior distributions of the original sample $x_{\text{orig}}$ and its augmented variants. That is, for $p_{\text{orig}} = \hat{p}(y \mid x_{\text{orig}})$, $p_{\text{augmix1}} = \hat{p}(y \mid x_{\text{augmix1}})$, $p_{\text{augmix2}} = \hat{p}(y \mid x_{\text{augmix2}})$, we replace the original loss $\mathcal{L}$ with the loss

$$\mathcal{L}(p_{\text{orig}}, y) + \lambda \, \text{JS}(p_{\text{orig}}; p_{\text{augmix1}}; p_{\text{augmix2}}). \quad (1)$$

To interpret this loss, imagine a sample from one of the three distributions $p_{\text{orig}}, p_{\text{augmix1}}, p_{\text{augmix2}}$. The Jensen-Shannon divergence can be understood to measure the average information that the sample reveals about the identity of the distribution from which it was sampled.

This loss can be computed by first obtaining $M = (p_{\text{orig}} + p_{\text{augmix1}} + p_{\text{augmix2}})/3$ and then computing

$$\text{JS}(p_{\text{orig}}; p_{\text{augmix1}}; p_{\text{augmix2}}) = \frac{1}{3}\Big(\text{KL}[p_{\text{orig}}\|M] + \text{KL}[p_{\text{augmix1}}\|M] + \text{KL}[p_{\text{augmix2}}\|M]\Big). \quad (2)$$

Unlike an arbitrary KL Divergence between $p_{\text{orig}}$ and $p_{\text{augmix}}$, the Jensen-Shannon divergence is upper bounded, in this case by the logarithm of the number of classes. Note that we could instead compute $\text{JS}(p_{\text{orig}}; p_{\text{augmix1}})$, though this does not perform as well. The gain of training with $\text{JS}(p_{\text{orig}}; p_{\text{augmix1}}; p_{\text{augmix2}}; p_{\text{augmix3}})$ is marginal. The Jensen-Shannon Consistency Loss impels to model to be stable, consistent, and insensitive across to a diverse range of inputs (Zheng et al., 2016; Kannan et al., 2018; Xie et al., 2019). Ablations are in Section 4.3 and Appendix A.

## 4 EXPERIMENTS

**Datasets.** The two CIFAR (Krizhevsky & Hinton, 2009) datasets contain small $32 \times 32 \times 3$ color natural images, both with 50,000 training images and 10,000 testing images. *CIFAR-10* has 10 categories, and *CIFAR-100* has 100. The *ImageNet* (Deng et al., 2009) dataset contains 1,000 classes of approximately 1.2 million large-scale color images.

In order to measure a model's resilience to data shift, we evaluate on the *CIFAR-10-C*, *CIFAR-100-C*, and *ImageNet-C* datasets (Hendrycks & Dietterich, 2019). These datasets are constructed by corrupting the original CIFAR and ImageNet test sets. For each dataset, there are a total of 15 noise, blur, weather, and digital corruption types, each appearing at 5 severity levels or intensities. Since these datasets are used to measure network behavior under data shift, we take care not to introduce these 15 corruptions into the training procedure.

The *CIFAR-10-P*, *CIFAR-100-P*, and *ImageNet-P* datasets also modify the original CIFAR and ImageNet datasets. These datasets contain smaller perturbations than CIFAR-C and are used to measure the classifier's prediction stability. Each example in these datasets is a video. For instance, a video with the brightness perturbation shows an image getting progressively brighter over time. We should like the network not to give inconsistent or volatile predictions between frames of the video as the brightness increases. Thus these datasets enable the measurement of the "jaggedness" (Azulay & Weiss, 2018) of a network's prediction stream.

**Metrics.** The *Clean Error* is the usual classification error on the clean or uncorrupted test data. In our experiments, corrupted test data appears at five different intensities or severity levels $1 \leq s \leq 5$. For a given corruption $c$, the error rate at corruption severity $s$ is $E_{c,s}$. We can compute the average error across these severities to create the unnormalized corruption error $\text{uCE}_c = \sum_{s=1}^{5} E_{c,s}$. On CIFAR-10-C and CIFAR-100-C we average these values over all 15 corruptions. Meanwhile, on ImageNet we follow the convention of normalizing the corruption error by the corruption error of AlexNet (Krizhevsky et al., 2012). We compute $\text{CE}_c = \sum_{s=1}^{5} E_{c,s} / \sum_{s=1}^{5} E_{c,s}^{\text{AlexNet}}$. The average of the 15 corruption errors $\text{CE}_{\text{Gaussian Noise}}, \text{CE}_{\text{Shot Noise}}, \ldots, \text{CE}_{\text{Pixelate}}, \text{CE}_{\text{JPEG}}$ gives us the *Mean Corruption Error (mCE)*.

Perturbation robustness is not measured by accuracy but whether video frame predictions match. Consequently we compute what is called the *flip probability*. Concretely, for videos such as those with steadily increasing brightness, we determine the probability that two adjacent frames, or two frames with slightly different brightness levels, have "flipped" or mismatched predictions. There are 10 different perturbation types, and the mean across these is the *mean Flip Probability (mFP)*. As with ImageNet-C, we can normalize by AlexNet's flip probabilities and obtain the *mean Flip Rate (mFR)*.

In order to assess a model's uncertainty estimates, we measure its miscalibration. Classifiers capable of reliably forecasting their accuracy are considered "calibrated." For instance, a calibrated classifier should be correct 70% of the time on examples to which it assigns 70% confidence. Let the classifier's confidence that its prediction $\hat{Y}$ is correct be written $C$. Then the idealized RMS Calibration Error is $\sqrt{\mathbb{E}_C[(\mathbb{P}(Y = \hat{Y}|C = c) - c)^2]}$, which is the squared difference between the accuracy at a given confidence level and actual the confidence level. In Appendix E, we show how to empirically estimate this quantity and calculate the Brier Score.

### 4.1 CIFAR-10 AND CIFAR-100

**Training Setup.** In the following experiments we show that AUGMIX endows robustness to various architectures including an All Convolutional Network (Springenberg et al., 2014; Salimans & Kingma, 2016), a DenseNet-BC ($k = 12, d = 100$) (Huang et al., 2017) , a 40-2 Wide ResNet (Zagoruyko & Komodakis, 2016), and a ResNeXt-29 ($32 \times 4$) (Xie et al., 2016). All networks use an initial learning rate of 0.1 which decays following a cosine learning rate (Loshchilov & Hutter, 2016). All input images are pre-processed with standard random left-right flipping and cropping prior to any augmentations. We do not change AUGMIX parameters across CIFAR-10 and CIFAR-100 experiments for consistency. The All Convolutional Network and Wide ResNet train for 100 epochs, and the DenseNet and ResNeXt require 200 epochs for convergence. We optimize with stochastic gradient descent using Nesterov momentum. Following Zhang et al. (2017); Guo et al. (2019), we use a weight decay of 0.0001 for Mixup and 0.0005 otherwise.

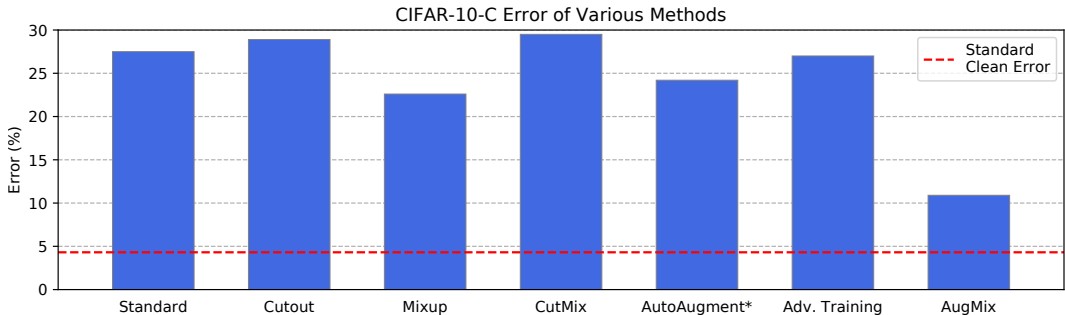

Figure 5: Error rates of various methods on CIFAR-10-C using a ResNeXt backbone. Observe that AUGMIX halves the error rate of prior methods and approaches the clean error rate.

|  |  | Standard | Cutout | Mixup | CutMix | AutoAugment* | Adv Training | AUGMIX |
|---|---|---|---|---|---|---|---|---|
| CIFAR-10-C | AllConvNet | 30.8 | 32.9 | 24.6 | 31.3 | 29.2 | 28.1 | **15.0** |
|  | DenseNet | 30.7 | 32.1 | 24.6 | 33.5 | 26.6 | 27.6 | **12.7** |
|  | WideResNet | 26.9 | 26.8 | 22.3 | 27.1 | 23.9 | 26.2 | **11.2** |
|  | ResNeXt | 27.5 | 28.9 | 22.6 | 29.5 | 24.2 | 27.0 | **10.9** |
| Mean |  | 29.0 | 30.2 | 23.5 | 30.3 | 26.0 | 27.2 | **12.5** |
| CIFAR-100-C | AllConvNet | 56.4 | 56.8 | 53.4 | 56.0 | 55.1 | 56.0 | **42.7** |
|  | DenseNet | 59.3 | 59.6 | 55.4 | 59.2 | 53.9 | 55.2 | **39.6** |
|  | WideResNet | 53.3 | 53.5 | 50.4 | 52.9 | 49.6 | 55.1 | **35.9** |
|  | ResNeXt | 53.4 | 54.6 | 51.4 | 54.1 | 51.3 | 54.4 | **34.9** |
| Mean |  | 55.6 | 56.1 | 52.6 | 55.5 | 52.5 | 55.2 | **38.3** |

Table 1: Average classification error as percentages. Across several architectures, AUGMIX obtains CIFAR-10-C and CIFAR-100-C corruption robustness that exceeds the previous state of the art.

**Results.** Simply mixing random augmentations and using the Jensen-Shannon loss substantially improves robustness and uncertainty estimates. Compared to the "Standard" data augmentation baseline ResNeXt on CIFAR-10-C, AUGMIX achieves 16.6% lower absolute corruption error as shown in Figure 5. In addition to surpassing numerous other data augmentation techniques, Table 1 demonstrates that these gains directly transfer across architectures and on CIFAR-100-C with zero additional tuning. Crucially, the robustness gains do not only exist when measured in aggregate. Figure 12 shows that AUGMIX improves corruption robustness across every individual corruption and severity level. Our method additionally achieves the lowest mFP on CIFAR-10-P across three different models all while maintaining accuracy on clean CIFAR-10, as shown in Figure 6 (left) and Table 6. Finally, we demonstrate that AUGMIX improves the RMS calibration error on CIFAR-10 and CIFAR-10-C, as shown in Figure 6 (right) and Table 5. Expanded CIFAR-10-P and calibration results are in Appendix D, and Fourier Sensitivity analysis is in Appendix B.

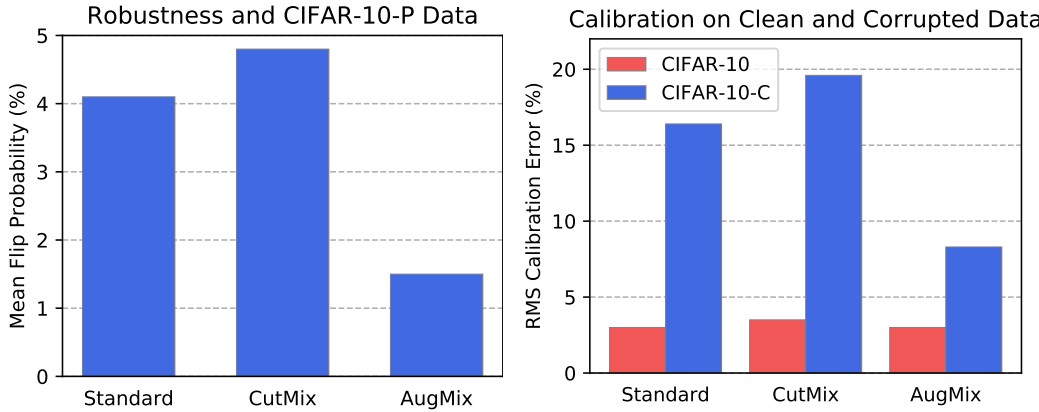

Figure 6: CIFAR-10-P prediction stability and Root Mean Square Calibration Error values for ResNeXt. AUGMIX simultaneously reduces flip probabilities and calibration error.

## 4.2 IMAGENET

**Baselines.** To demonstrate the utility of AUGMIX on ImageNet, we compare to many techniques designed for large-scale images. While techniques such as Cutout (Devries & Taylor, 2017) have not been demonstrated to help on the ImageNet scale, and while few have had success training adversarially robust models on ImageNet (Engstrom et al., 2018), other techniques such as Stylized ImageNet have been demonstrated to help on ImageNet-C. Patch Uniform (Lopes et al., 2019) is similar to Cutout except that randomly chosen regions of the image are injected with uniform noise; the original paper uses Gaussian noise, but that appears in the ImageNet-C test set so we use uniform noise. We tune Patch Uniform over 30 hyperparameter settings. Next, AutoAugment (Cubuk et al., 2018) searches over data augmentation policies to find a high-performing data augmentation policy. We denote AutoAugment results with AutoAugment* since we remove augmentation operations that overlap with ImageNet-C corruptions, as with AUGMIX. We also test with Random AutoAugment*, an augmentation scheme where each image has a randomly sampled augmentation policy using AutoAugment* operations. In contrast to AutoAugment, Random AutoAugment* and AUGMIX require far less computation and provide more augmentation variety, which can offset their lack of optimization. Note that Random AutoAugment* is different from RandAugment introduced recently by Cubuk et al. (2019): RandAugment uses AutoAugment operations and optimizes a single distortion magnitude hyperparameter for all operations, while Random AutoAugment* randomly samples magnitudes for each operation and uses the same operations as AUGMIX. MaxBlur Pooling (Zhang, 2019) is a recently proposed architectural modification which smooths the results of pooling. Now, Stylized ImageNet (SIN) is a technique where models are trained with the original ImageNet images and also ImageNet images with style transfer applied. Whereas the original Stylized ImageNet technique pretrains on ImageNet-C and performs style transfer with a content loss coefficient of $0$ and a style loss coefficient of $1$, we find that using $0.5$ content and style loss coefficients decreases the mCE by 0.6%. Later, we show that SIN and AUGMIX can be combined. All models are trained from scratch, except MaxBlur Pooling models which has trained models available.

**Training Setup.** Methods are trained with ResNet-50 and we follow the standard training scheme of Goyal et al. (2017), in which we linearly scale the learning rate with the batch size, and use a learning rate warm-up for the first 5 epochs, and AutoAugment and AUGMIX train for 180 epochs. All input images are first pre-processed with standard random cropping horizontal mirroring.

| | | Noise | | | Blur | | | | Weather | | | | Digital | | | | |
|---|---|---|---|---|---|---|---|---|---|---|---|---|---|---|---|---|---|
| Network | Clean | Gauss. | Shot | Impulse | Defocus | Glass | Motion | Zoom | Snow | Frost | Fog | Bright | Contrast | Elastic | Pixel | JPEG | **mCE** |
| Standard | 23.9 | 79 | 80 | 82 | 82 | 90 | 84 | 80 | 86 | 81 | 75 | 65 | 79 | 91 | 77 | 80 | 80.6 |
| Patch Uniform | 24.5 | 67 | 68 | 70 | 74 | 83 | 81 | 77 | 80 | 74 | 75 | 62 | 77 | 84 | 71 | 71 | 74.3 |
| AutoAugment* (AA) | 22.8 | 69 | 68 | 72 | 77 | 83 | 80 | 81 | 79 | 75 | 64 | **57** | 70 | 88 | **57** | 71 | 72.7 |
| Random AA* | 23.6 | 70 | 71 | 72 | 80 | 86 | 82 | 81 | 81 | 77 | 72 | 61 | 75 | 88 | 73 | 72 | 76.1 |
| MaxBlur pool | 23.0 | 73 | 74 | 76 | 74 | 86 | 78 | 77 | 77 | 72 | **63** | 56 | 68 | 86 | 71 | 71 | 73.4 |
| SIN | 27.2 | 69 | 70 | 70 | 77 | 84 | 76 | 82 | 74 | 75 | 69 | 65 | 69 | 80 | 64 | 77 | 73.3 |
| AUGMIX | **22.4** | 65 | 66 | 67 | 70 | 80 | 66 | **66** | 75 | 72 | 67 | 58 | 58 | 79 | 69 | 69 | 68.4 |
| AUGMIX+SIN | 25.2 | **61** | **62** | **61** | **69** | **77** | **63** | 72 | **66** | **68** | **63** | 59 | **52** | **74** | **60** | **67** | **64.9** |

Table 2: Clean Error, Corruption Error (CE), and mCE values for various methods on ImageNet-C. The mCE value is computed by averaging across all 15 CE values. AUGMIX reduces corruption error while improving clean accuracy, and it can be combined with SIN for greater corruption robustness.

**Results.** Our method achieves 68.4% mCE as shown in Table 2, down from the baseline 80.6% mCE. Additionally, we note that AUGMIX allows straightforward stacking with other methods such as SIN to achieve an even lower corruption error of 64.1% mCE. Other techniques such as AutoAugment* require much tuning, while ours does not. Across increasing severities of corruptions, our method also produces much more calibrated predictions measured by both the Brier Score and RMS Calibration Error as shown in Figure 7. As shown in Table 3, AUGMIX also achieves a state-of-the art result on ImageNet-P at with an mFR of 37.4%, down from 57.2%. We demonstrate that scaling up AUGMIX from CIFAR to ImageNet also leads to state-of-the-art results in robustness and uncertainty estimation.

### 4.3 ABLATIONS

We locate the utility of AUGMIX in three factors: training set diversity, our Jensen-Shannon divergence consistency loss, and mixing. Improving training set diversity via increased variety of augmentations can greatly improve robustness. For instance, augmenting each example with a

| Network | Clean | Noise | | Blur | | Weather | | Digital | | | | mFR |
| | | Gaussian | Shot | Motion | Zoom | Snow | Bright | Translate | Rotate | Tilt | Scale | |
|---|---|---|---|---|---|---|---|---|---|---|---|---|
| Standard | 23.9 | 57 | 55 | 62 | 65 | 66 | 65 | 43 | 53 | 57 | 49 | 57.2 |
| Patch Uniform | 24.5 | **32** | **25** | 50 | 52 | 54 | 57 | 40 | 48 | 49 | 46 | 45.3 |
| AutoAugment* (AA) | 22.8 | 50 | 45 | 57 | 68 | 63 | 53 | 40 | 44 | 50 | 46 | 51.7 |
| Random AA* | 23.6 | 53 | 46 | 53 | 63 | 59 | 57 | 42 | 48 | 54 | 47 | 52.2 |
| SIN | 27.2 | 53 | 50 | 57 | 72 | 51 | 62 | 43 | 53 | 57 | 53 | 55.0 |
| MaxBlur pool | 23.0 | 52 | 51 | 59 | 63 | 57 | 64 | 34 | 43 | 49 | 40 | 51.2 |
| AUGMIX | **22.4** | 46 | 41 | **30** | **47** | 38 | **46** | **25** | **32** | **35** | **33** | **37.4** |
| AUGMIX+SIN | 25.2 | 45 | 40 | **30** | 54 | **32** | 48 | 27 | 35 | 38 | 39 | 38.9 |

Table 3: ImageNet-P results. The mean flipping rate is the average of the flipping rates across all 10 perturbation types. AUGMIX improves perturbation stability by approximately 20%.

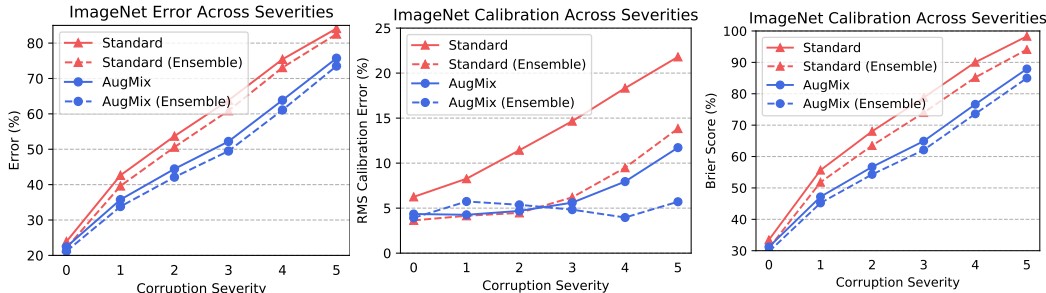

Figure 7: Uncertainty results on ImageNet-C. Observe that under severe data shifts, the RMS calibration error with ensembles and AUGMIX is remarkably steady. Even though classification error increases, calibration is roughly preserved. Severity zero denotes clean data.

randomly sampled augmentation chain decreases the error rate of Wide ResNet on CIFAR-10-C from 26.9% to 17.0% Table 4. Adding in the Jensen-Shannon divergence consistency loss drops error rate further to 14.7%. Mixing random augmentations without the Jensen-Shannon divergence loss gives us an error rate of 13.1%. Finally, re-introducing the Jensen-Shannon divergence gives us AUGMIX with an error rate of 11.2%. Note that adding even more mixing is not necessarily beneficial. For instance, applying AUGMIX on top of Mixup increases the error rate to 13.3%, possibly due to an increased chance of manifold intrusion (Guo et al., 2019). Hence AUGMIX's careful combination of variety, consistency loss, and mixing explain its performance.

| Method | CIFAR-10-C Error Rate | CIFAR-100-C Error Rate |
|---|---|---|
| Standard | 26.9 | 53.3 |
| AutoAugment* | 23.9 | 49.6 |
| Random AutoAugment* | 17.0 | 43.6 |
| Random AutoAugment* + JSD Loss | 14.7 | 40.8 |
| AugmentAndMix (No JSD Loss) | 13.1 | 39.8 |
| AUGMIX (Mixing + JSD Loss) | 11.2 | 35.9 |

Table 4: Ablating components of AUGMIX on CIFAR-10-C and CIFAR-100-C. Variety through randomness, the Jensen-Shannon divergence (JSD) loss, and augmentation mixing confer robustness.

## 5 CONCLUSION

AUGMIX is a data processing technique which mixes randomly generated augmentations and uses a Jensen-Shannon loss to enforce consistency. Our simple-to-implement technique obtains state-of-the-art performance on CIFAR-10/100-C, ImageNet-C, CIFAR-10/100-P, and ImageNet-P. AUGMIX models achieve state-of-the-art calibration and can maintain calibration even as the distribution shifts. We hope that AUGMIX will enable more reliable models, a necessity for models deployed in safety-critical environments.

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

## A    HYPERPARAMETER ABLATIONS

In this section we demonstrate that AUGMIX's hyperparameters are not highly sensitive, so that AUGMIX performs reliably without careful tuning. For this set of experiments, the baseline AUGMIX model trains for 90 epochs, has a mixing coefficient of $\alpha = 0.5$, has 3 examples per Jensen-Shannon Divergence (1 clean image, 2 augmented images), has a chain depth stochastically varying from 1 to 3, and has $k = 3$ augmentation chains. Figure 8 shows that the performance of various AUGMIX models with different hyperparameters. Under these hyperparameter changes, the mCE does not change substantially.

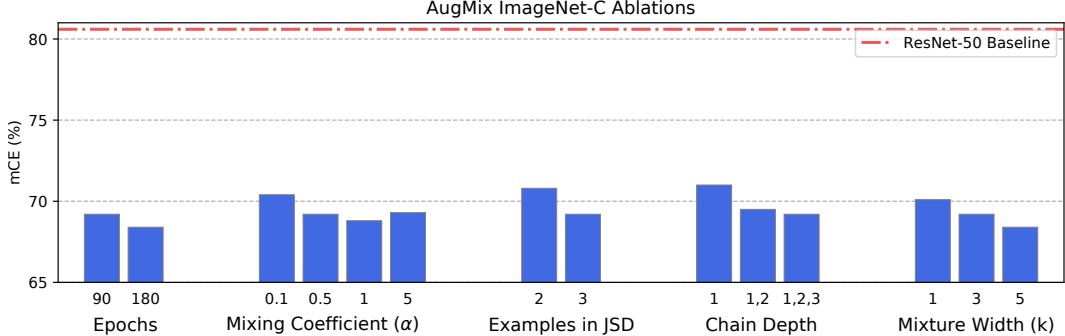

Figure 8: AUGMIX hyperparameter ablations on ImageNet-C. ImageNet-C classification performance is stable changes to AUGMIX's hyperparameters.

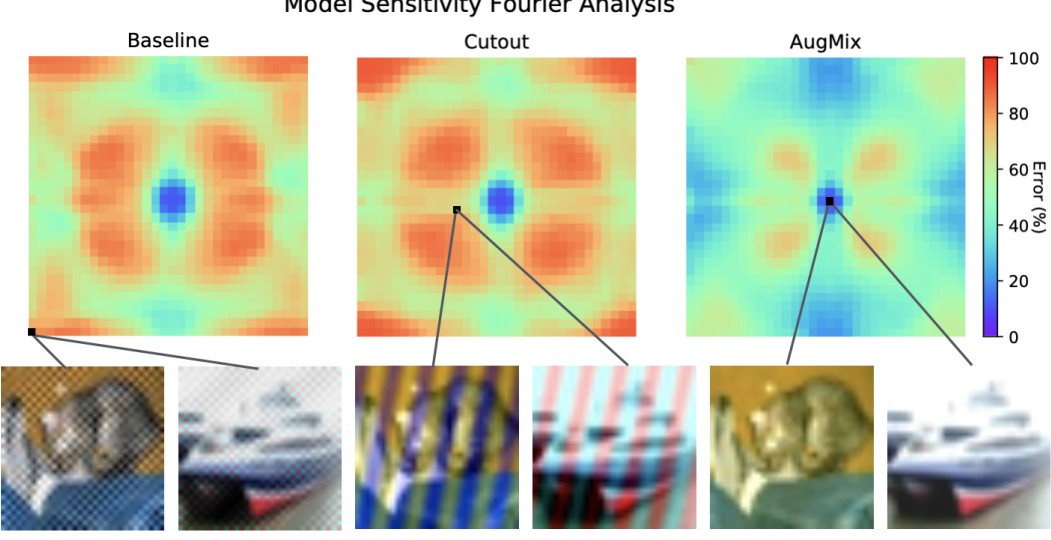

Figure 9: Fourier Sensitivity Heatmap of the baseline Wide ResNet, Cutout, and AUGMIX CIFAR-10 models. All Fourier basis perturbations are added to clean CIFAR-10 test images. AUGMIX maintains robustness at low frequencies and is far more robust to mid and high frequency modifications. Example perturbed images are shown above, with black pointer lines indicating the Fourier basis vector used to perturb the image. For each basis vector we compute the error rate of the model after perturbing the entire test set.

## B    FOURIER ANALYSIS

A commonly mentioned hypothesis (Gilmer & Hendrycks, 2019) for the lack of robustness of deep neural networks is that they readily latch onto spurious high-frequency correlations that exist in the data. In order to better understand the reliance of models to such correlations, we measure model sensitivity to additive noise at differing frequencies. We create a $32 \times 32$ sensitivity heatmap. That is,

we add a total of $32 \times 32$ Fourier basis vectors to the CIFAR-10 test set, one at a time, and record the resulting error rate after adding each Fourier basis vector. Each point in the heatmap shows the error rate on the CIFAR-10 test set after it has been perturbed by a single Fourier basis vector. Points corresponding to low frequency vectors are shown in the center of the heatmap, whereas high frequency vectors are farther from the center. For further details on Fourier sensitivity analysis, we refer the reader to Section 2 of Yin et al. (2019). In Figure 9 we observe that the baseline model is robust to low frequency perturbations but severely lacks robustness to high frequency perturbations, where error rates exceed 80%. The model trained with Cutout shows a similar lack of robustness. In contrast, the model trained with AUGMIX maintains robustness to low frequency perturbations, and on the mid and high frequencies AUGMIX is conspicuously more robust.

## C  AUGMENTATION OPERATIONS

The augmentation operations we use for AUGMIX are shown in Figure 10.

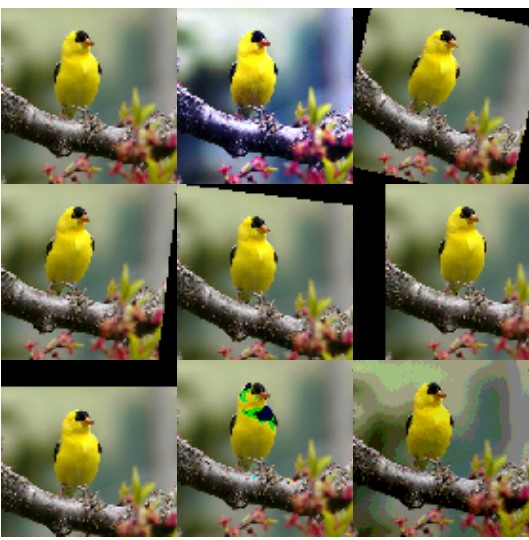

Figure 10: Illustration of augmentation operations applied to the same image. Some operation severities have been increased to show detail.

We do not use augmentations such as `contrast`, `color`, `brightness`, `sharpness`, and `Cutout` as they may overlap with ImageNet-C test set corruptions. We should note that augmentation choice requires additional care. Guo et al. (2019) show that blithely applying augmentations can potentially cause augmented images to take different classes. Figure 11 shows how histogram color swapping augmentation may change a bird's class, leading to a manifold intrusion.

Manifold Intrusion from Color Augmentation

Original                    Color Augmented

Figure 11: An illustration of manifold intrusion (Guo et al., 2019), where histogram color augmentation can change the image's class.

# D  ADDITIONAL RESULTS

We include various additional results for CIFAR-10, CIFAR-10-C and CIFAR-10-P below. Figure 12 reports accuracy for each corruption, Table 5 reports calibration results for various architectures and Table 6 reports clean error and mFR. We refer to Section 4.1 for details about the architecture and training setup.

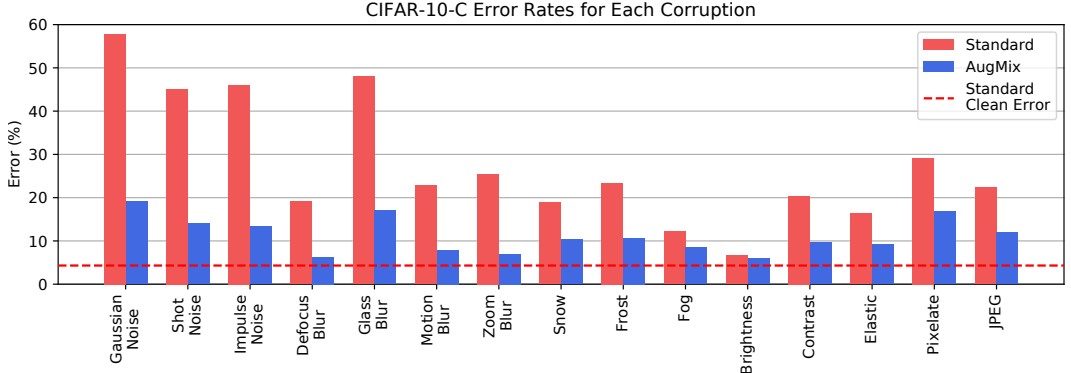

Figure 12: AUGMIX improves corruption robustness across all CIFAR-10-C noise, blur, weather, and digital corruptions, despite the model never having seen these corruptions during training.

| | | Standard | Cutout | Mixup | CutMix | AutoAugment* | Adv Training | AUGMIX |
|---|---|---|---|---|---|---|---|---|
| CIFAR-10 | AllConvNet | 5.4 | 4.0 | 12.6 | 3.1 | 4.2 | 11.1 | 2.2 |
| | DenseNet | 7.5 | 6.4 | 15.6 | 5.4 | 6.0 | 16.2 | 5.0 |
| | WideResNet | 6.8 | 3.8 | 14.0 | 5.0 | 4.7 | 10.7 | 4.2 |
| | ResNeXt | 3.0 | 4.4 | 13.5 | 3.5 | 3.3 | 5.8 | 3.0 |
| Mean | | 5.7 | 4.7 | 13.9 | 4.2 | 4.6 | 11.0 | 3.6 |
| CIFAR-10-C | AllConvNet | 21.2 | 21.3 | 9.7 | 15.4 | 16.2 | 10.4 | 5.2 |
| | DenseNet | 26.7 | 27.8 | 12.9 | 25.6 | 21.1 | 15.0 | 11.7 |
| | WideResNet | 27.6 | 19.6 | 11.1 | 17.8 | 17.1 | 10.6 | 8.7 |
| | ResNeXt | 16.4 | 21.4 | 11.7 | 19.6 | 15.1 | 11.6 | 8.3 |
| Mean | | 23.0 | 22.5 | 11.4 | 19.6 | 17.4 | 11.9 | 8.5 |

Table 5: RMS Calibration Error of various models and data augmentation methods across CIFAR-10 and CIFAR-10-C. All values are reported as percentages.

| | | Standard | Cutout | Mixup | CutMix | AutoAugment* | Adv Training | AUGMIX |
|---|---|---|---|---|---|---|---|---|
| CIFAR-10 | AllConvNet | 6.1 | 6.1 | 6.3 | 6.4 | 6.6 | 18.9 | 6.5 |
| | DenseNet | 5.8 | 4.8 | 5.5 | 5.3 | 4.8 | 17.9 | 4.9 |
| | WideResNet | 5.2 | 4.4 | 4.9 | 4.6 | 4.8 | 17.1 | 4.9 |
| | ResNeXt | 4.3 | 4.4 | 4.2 | 3.9 | 3.8 | 15.4 | 4.2 |
| Mean | | 5.4 | 4.9 | 5.2 | 5.0 | 5.0 | 17.3 | 5.1 |
| CIFAR-10-P | AllConvNet | 4.2 | 5.0 | 3.9 | 4.5 | 4.0 | 2.0 | 1.5 |
| | DenseNet | 5.0 | 5.7 | 3.9 | 6.3 | 4.8 | 2.1 | 1.8 |
| | WideResNet | 4.2 | 4.3 | 3.4 | 4.6 | 4.2 | 2.2 | 1.6 |
| | ResNeXt | 4.0 | 4.5 | 3.2 | 5.2 | 4.2 | 2.5 | 1.5 |
| Mean | | 4.3 | 4.9 | 3.6 | 5.2 | 4.3 | 2.2 | 1.6 |

Table 6: CIFAR-10 Clean Error and CIFAR-10-P mean Flip Probability. All values are percentages. While adversarial training performs well on CIFAR-10-P, it induces a substantial drop in accuracy (increase in error) on clean CIFAR-10 where AUGMIX does not.

# E    CALIBRATION METRICS

Due to the finite size of empirical test sets, the RMS Calibration Error must be estimated by partitioning all $n$ test set examples into $b$ contiguous bins $\{B_1, B_2, \ldots, B_b\}$ ordered by prediction confidence. In this work we use bins which contain 100 predictions, so that we adaptively partition confidence scores on the interval $[0, 1]$ (Nguyen & O'Connor, 2015; Hendrycks et al., 2019b). Other works partition the interval $[0, 1]$ with 15 bins of uniform length (Guo et al., 2017). With these $b$ bins, we estimate the *RMS Calibration Error* empirically with the formula

$$\sqrt{\sum_{i=1}^{b} \frac{|B_i|}{n} \left( \frac{1}{|B_i|} \sum_{k \in B_i} \mathbb{1}(y_k = \hat{y}_k) - \frac{1}{|B_i|} \sum_{k \in B_i} c_k \right)^2}. \tag{3}$$

This is separate from classification error because a random classifier with an approximately uniform posterior distribution is approximately calibrated. Also note that adding the "refinement" $\mathbb{E}_C[(\mathbb{P}(Y = \hat{Y}|C = c)(1 - (\mathbb{P}(Y = \hat{Y}|C = c))]$ to the square of the RMS Calibration Error gives us the *Brier Score* (Nguyen & O'Connor, 2015).

