# OpenReview forum: "AugMix: A Simple Data Processing Method to Improve Robustness and Uncertainty"
_ICLR.cc/2020/Conference — Accept (Poster)_

### Official Review · AnonReviewer2 · 2019-10-20
**Official Blind Review #2**

**Rating:** 8

**Review:**

This paper proposes a method called AugMix, which is intended to improve model robustness to data distribution shift. AugMix appears fairly simple to implement. Several new images are created by augmenting an original image through chains of sequentially applied transformations (the "Aug" part of AugMix), then the augmented images are combined together, along with the original image, via a weighted sum (the "Mix" part of AugMix). Additionally, a Jensen-Shannon Divergence consistency loss is applied during training to encourage the model to make similar predictions for all augmented variations of a single image. This technique is shown to achieve state-of-the-art performance on standard robustness benchmarks without loss of clean test accuracy, and is also shown to improve calibration of model confidence estimates.

Overall, I would tend to vote for accepting this paper. The method is simple yet effective, the paper is very well written and easy to follow, and experiments are extensive and, for the most part, convincing.

Questions:
1) The one main concern I have with the training procedure is the amount of time the models were trained for. It is known that model trained with aggressive data augmentation schemes often require much longer training than normal in order to fully benefit from the stronger augmentations. For example, AutoAugment trains ImageNet models for 270 epochs [1], while CutMix trains for 300 epochs [2]. However, the ImageNet experiments in this paper claim to follow the procedure outlined in [3], which only trains for 90 epochs. This is reflected in the clean test accuracy, where AutoAugment only appears to provide a 0.3% gain over standard training, while we might expect 1.3% improvement (according to [2]). The AllConvNet and WideResNet in CIFAR-10 and CIFAR-100 experiments were also trained for only 100 epochs each, where 200 is more conventional. Again this shows in the reported numbers: on WideResNet for CIFAR-10, Mixup only has a 0.3% gain were as we might expect 1% improvement instead [4], and AutoAugment has 0.4% improvement, were as we might expect 1.3% gain if trained longer [1]. My question then is, how much does training time affect results? Do AugMix, and other techniques such as Mixup, CutMix, and AutoAugment, achieve better robustness when models are trained for longer, or do they become more brittle as training time is extended?

2) For the Jensen-Shannon divergence consistency, how much worse does it perform when using JS(Porig;Paugmix1) versus JS(Porig;Paugmix1;Paugmix2)? What might cause this behaviour?

3) Patch Gaussian is changed to Patch Uniform to avoid overlap with corruptions in ImageNet-C. How does Patch Uniform compare to Patch Gaussian in terms of performance for non-Gaussian noise corruptions?

4) How does AugMix perform as an augmentation technique in terms of clean test accuracy compared to other SOTA techniques? Is there a trade-off between clean test accuracy and robustness, or does AugMix improve performance in both domains? Can AugMix be combined with other augmentation techniques or does this destroy robustness properties?

Things to improve the paper that did not impact the score:
5) It would be nice if the best result in each column could be bolded in Tables 2-4.

References:
[1] Cubuk, Ekin D., Barret Zoph, Dandelion Mane, Vijay Vasudevan, and Quoc V. Le. "Autoaugment: Learning augmentation policies from data." CVPR (2019).

[2] Yun, Sangdoo, Dongyoon Han, Seong Joon Oh, Sanghyuk Chun, Junsuk Choe, and Youngjoon Yoo. "Cutmix: Regularization strategy to train strong classifiers with localizable features." ICCV (2019).

[3] Goyal, Priya, Piotr Dollár, Ross Girshick, Pieter Noordhuis, Lukasz Wesolowski, Aapo Kyrola, Andrew Tulloch, Yangqing Jia, and Kaiming He. "Accurate, large minibatch sgd: Training imagenet in 1 hour." arXiv preprint arXiv:1706.02677 (2017).

[4] Zhang, Hongyi, Moustapha Cisse, Yann N. Dauphin, and David Lopez-Paz. "mixup: Beyond empirical risk minimization." ICLR (2018).

**Experience Assessment:**

I have read many papers in this area.

**Review Assessment: Checking Correctness Of Derivations And Theory:**

N/A

**Review Assessment: Checking Correctness Of Experiments:**

I carefully checked the experiments.

**Review Assessment: Thoroughness In Paper Reading:**

I read the paper thoroughly.

---

> ### Author Response · Authors · 2019-11-11
> **Reviewer #2 Reply**
>
> Thank you for your detailed review. We are glad you liked the method and hope that you will champion our paper.
>
> 1. We trained the AllConvNet and Wide ResNet for 100 epochs since we used a cosine learning rate schedule and not a waterfall learning rate schedule; the latter schedule usually requires 200 epochs, but the former requires fewer epochs for these architectures. On CIFAR-10-C, we observed that training a Wide ResNet for 200 epochs instead of 100 decreased the AugMix error rate by ~0.5%, so AugMix can provide some amount of additional training robustness when trained for longer. However, training with Cutmix for 200 epochs on CIFAR-10-C increases the error rate by 2%. You are correct to note that our AutoAugment run was trained for 90 epochs on ImageNet, and we have updated the results with a 180 epoch run. 180 epochs gives similar performance to 270 epochs, according to a recent correspondence with the authors of AutoAugment. The clean accuracy of AutoAugment is nearly that of the accuracy in the original paper, even though we disable a few operations so as to maintain separation from ImageNet-C test corruptions.
>
> 2. We have added new hyperparameter analysis experiments in Appendix A. This section includes an experiment to analyze the Jensen-Shannon loss. One possible explanation for the performance gain realized by JS(Porig;Paugmix1;Paugmix2) over JS(Porig;Paugmix1) is that the former reduces the variance of the estimate of the true mixture distribution.
>
> 3. It is difficult to directly compare Patch Uniform to numbers from the Patch Gaussian paper since we follow convention and evaluate on 224x224 images not 299x299 images, while the Patch Gaussian paper evaluates on 299x299 images. In a footnote on page 4, the authors note that they will update their paper with results on 224x224 images, after which we will be able to directly compare against a well-tuned tuned form of Patch Gaussian.
>
> 4. We note in Table 2 that AugMix also improves clean accuracy on ImageNet, though we are primarily interested in improving robustness. While it is plausible that this field may encounter a (possibly small) tradeoff between robustness and accuracy, our simultaneous improvements in both directions show that we are not at that point yet for corruption and perturbation robustness. As we limited ourselves to the set of augmentation operators used in AutoAugment, expanding the pool of label-preserving data augmentations would be a straightforward extension of AugMix that would likely yield additional improvement. Our experiments on AugMix+SIN show that AugMix may be combined in alongside other robustness methods without additional tuning.
>
> 5. Your suggestion is incorporated in the new version of the paper. Thank you.

---

> > ### Comment · AnonReviewer2 · 2019-11-13
> > **Reply**
> >
> > Thank you for running the additional experiments. My concerns have been  addressed.

---

### Official Review · AnonReviewer3 · 2019-10-21
**Official Blind Review #3**

**Rating:** 3

**Review:**

The paper discusses a new  data augmentation method which improves the accuracy of the network for several specific shifted domain scenarios. The main goal of the paper is to increase the robustness of the deep model trained on the augmented data to generalize well beyond the data corruption like the  rotation, translation, noise,.... For each input, they apply  $k$ different operation of image shift and make the weighted combination of them. The weight vector is generated randomly from Dirichlet distribution with the parameter $\alpha$.  The weighted combined images would be added to the original image in convex combination. The convex weights are generated from distribution Beta with parameter $\beta$. Later they train the network with adding the Jensen-Shannon divergence for the posterior distributions of augmented images as the consistency regularizer.  They show this data augmentation will increase the accuracy of the model for shifted and non-shifted domains and also it leads to more calibrated model for domain shift problem.

Pros:
the paper is well-written with clear implementation details. The level of experiments are wide and cover different aspects. The experiments shows the significant improvement compared to several baselines. The authors conducted the experiments for a wide range of model-datasets to show the validity of their ideas.

Cons:
1- The title of this work is a strong claim that is not supported in the paper. In this paper, it is mentioned that AugMix is a data augmentation method that generates data to add to the training set and after training with data augmentation, the method would be more robust to other distortions that can be added to the datasets. Generally, the definition of domain shift is wider than just adding perturbation to the dataset.  To support the claim, the paper should also report the results for similar tasks datasets such as CIFAT10-STL10- or MINIST-SVHN for different models and with different domain adaptation methods. The claim about the improvement of uncertainty also is not supported well by the experiments. The method should be tested for many model-datasets specifically, to support improving the  uncertainty under the domain shift idea like the paper [1].

2- The novelty of the work is limited. The generating method of distorted  images is the combination of previously proposed methods like [2] and [3].  The motivation of why the proposed method is working well is not clear. How this objective function can improve the robustness to the image perturbation but it does not lose the accuracy is not discussed. It would be better if the proposed method were supported by theory and also the intuition and explained why it should get better results than previous data augmentation methods such as AutoAugment [3].

3-  Fine-tuning the parameters like $k$, $\alpha$ and $\beta$ is not discussed at all.

4- To show the robustness of the proposed method to domain shift, the paper compares the proposed method to other data augmentation methods that are not designed for domain shift which seems unfair.

References:
[1] Ovadia, Yaniv, et al. "Can You Trust Your Model's Uncertainty? Evaluating Predictive Uncertainty Under Dataset Shift." arXiv preprint arXiv:1906.02530 (2019).
[2] Zhang, Hongyi, et al. "mixup: Beyond empirical risk minimization." arXiv preprint arXiv:1710.09412 (2017).
[3] Cubuk, Ekin D., et al. "Autoaugment: Learning augmentation policies from data." arXiv preprint arXiv:1805.09501 (2018).


**Experience Assessment:**

I do not know much about this area.

**Review Assessment: Checking Correctness Of Derivations And Theory:**

N/A

**Review Assessment: Checking Correctness Of Experiments:**

I carefully checked the experiments.

**Review Assessment: Thoroughness In Paper Reading:**

I read the paper at least twice and used my best judgement in assessing the paper.

---

> ### Author Response · Authors · 2019-11-09
> **Clarifying the Problem Setup and a Comparison to AutoAugment**
>
> Thank you for your detailed response.
>
> 1. “The claim about the improvement of uncertainty also is not supported well by the experiments”
> We should like to point to the middle of Figure 7 showing calibration on ImageNet-C. This is a challenging problem as pointed out by the paper you mentioned [1]. AugMix significantly improves the calibration of the baseline method. Furthermore, combining AugMix with ensembles (the best performing method in Ovadia et al. [1]) significantly improves performance and achieves much better calibration under distributional skew as demonstrated by the near-horizontal calibration error line. To the best of our knowledge, AugMix + ensembles achieves state-of-the-art performance on calibration under distribution skew on ImageNet-C. If ensembles are too expensive, then a single-model with AugMix provides superior ImageNet-C calibration over ensembles. In addition, Figure 6 (right) and Table 5 also show that AugMix significantly improves calibration. We hope this evidence substantiates our claim that AugMix improves uncertainty estimates.
>
> “Data shift” is sometimes used interchangeably with “distributional skew” and “distribution shift.” For instance, the paper you mentioned by Ovadia et al. [1] evaluate “calibration under dataset shift” on images using ImageNet-C and CIFAR-10-C, and we do too.
>
> That said, “data shift” is not often used interchangeably with “domain adaptation.” Our paper does not contain experiments with MNIST classifiers transferring to SVHN since that is in the realm of domain adaptation, a setup which assumes knowledge of the structure of the data shift or access to a fine-tuning set. We consider the problem of robustness to unseen corruptions, where we assume no knowledge of the data shift a priori.
>
> 2. Comparing AugMix to AutoAugment and Mixup
> We believe that the simplicity of our method is a feature. AugMix is not a direct combination of AutoAugment and Mixup. AutoAugment requires training several thousand models to find an augmentation policy, whereas AugMix requires training only one. Hence, its computational cost is in league with that of traditional data augmentation techniques, but even so AugMix can outperform AutoAugment. While we use convex combinations of augmentations of one image, this does not make it an extension of Mixup. In Mixup, examples are from different classes are mixed, while we do nothing of the sort. While AugMix's name may suggest that it is a combination of AutoAugment and Mixup, the proposed method does not mix different training images and obviates the need for training several thousand models.
>
> We believe AugMix works better because (i) augmentations produced by AugMix are more "diverse" as the base operations are randomly sampled and randomly mixed in every minibatch and (ii) consistency between augmentations is enforced with our Jensen-Shannon divergence loss. Our ablation experiments in Table 4 show the relative contributions of these ingredients.
>
> 3. Further ablations
> Thanks to your reasonable request, we are running numerous additional ablations. We aim to share these results soon, which are so far indicating that AugMix is stable across different hyperparameter choices.
>
> 4. Competing with other techniques
> There are few techniques in the nascent area of data shift. However, we compared to all existing techniques proposed to tackle data shift (SIN, MaxBlurPool, etc.). In addition, we also compared to numerous other techniques (Cutmix, adversarial training, etc.) in order to provide an extensive comparison.
>
> We hope we were able to address your concerns and we thank you for your helpful suggestions. Do you have any remaining concerns?
>
> [1] Ovadia, Yaniv, et al. "Can You Trust Your Model's Uncertainty? Evaluating Predictive Uncertainty Under Dataset Shift." NeurIPS 2019.

---

> > ### Comment · AnonReviewer3 · 2019-11-13
> > **My concerns are addressed**
> >
> > Thank you for the clarification and adding hyperparameter-tunning results.  Most of my concerns are resolved.  But I suggest to clearly address the assumption of your paper, the robust and calibrated solution for unseen corruptions.

---

> > > ### Author Response · Authors · 2019-11-13
> > > **"Data Shift" Removed, Addressing Concern**
> > >
> > > Reviewer 3, thank you for your reply.
> > >
> > > We followed previous works in characterizing corruption and perturbation robustness as robustness to data shift, but we have heeded your valid concern and have now more clearly specified that we are considering robustness and uncertainty on corrupted distributions that are unseen until test time.
> > > Specifically, we have changed the title by removing the phrase "data shift" altogether from the title. This change is reflected in the PDF, but not yet on the OpenReview website due to OpenReview's restrictions. Should this paper be accepted, we should be able to edit the title and abstract on OpenReview as well. In the text of the PDF, we now more heavily emphasize that we are considering unseen corruption and perturbations. The title, introduction, abstract, and conclusion have been modified to conform to your recommendation.
> > >
> > > Thank you for your responsiveness.

---

> > > ### Author Response · Authors · 2019-11-15
> > > **Double Check**
> > >
> > > Reviewer 3,
> > >
> > > Thank you for your responsiveness throughout this week and helping us improve our paper.
> > >
> > > We have added a new Appendix B to give more analysis into how AugMix works.
> > > After tomorrow OpenReview will not allow us to answer further questions or address other concerns.
> > > Do you have any further experiments you should like us to run for you or have any other questions?

---

### Official Review · AnonReviewer1 · 2019-10-22
**Official Blind Review #1**

**Rating:** 8

**Review:**

The paper proposes a novel method called augMix, which creates synthetic samples by mixing multiple augmented images. Coupled with a Jensen-Shannon Divergence consistency loss, the proposed method has been experimentally, using CIFAR10, CIFAR100, and ImageNet, shown to be able to improve over some augmentation methods in terms of robustness and uncertainty.

The paper is very well written and easy to follow. The idea of the approach is simple, and should be easy to be implemented. The evaluation of the proposed method is currently based on experimental evidence. Nevertheless, the empirical studies in its current form could be further improved. Please see my detailed comments below.

1. The proposed approach relies on a chain of augmented methods. In this sense, experimental studies on the sensitivity for how the augmentation methods in the chain (e.g., augmentation operations) and their chain structure (e.g., length of the chain) impact the performance of the augMix should be provided. This is in particular relevant because the authors did mention that “adding even more mixing is not necessarily beneficial” on page 8.

2. Since the proposed method mixes multiple augmented images, a more appropriate comparison baseline would be a method involving creating synthetic data with multiple images. For example, the n-fold Mixup method as discussed in Guo AAAI2019 (Mixup as Locally Linear Out-Of-Manifold Regularization).

3. Some experimental results/observations deserve further discussions. For example, on page 8, the authors mention that “applying augMix on top of Mixup increases the error rate to 13.3%”. I wonder if the authors could provide any insights or hypothesis on why the proposed model behaviors in this way?

4. Would that be any manifold intrusion issue as discussed in Guo’s AAAI2019 paper? That is, would it be possible to generate images that are very close to some real images but with different labels? For example, by looking at the bottom-center image in the Appendix B, the synthetic image created seems to be close to some birds with other categories.

5. Does the method work for other network architectures such as DenseNet?


*********new  comment**********
During the rebuttal period, the paper has been improved with additional experimental results, analyses, and observations. I therefore have adjusted my evaluation score accordingly.



**Experience Assessment:**

I have published one or two papers in this area.

**Review Assessment: Checking Correctness Of Derivations And Theory:**

I carefully checked the derivations and theory.

**Review Assessment: Checking Correctness Of Experiments:**

I carefully checked the experiments.

**Review Assessment: Thoroughness In Paper Reading:**

I read the paper thoroughly.

---

> ### Author Response · Authors · 2019-11-08
> **AugMix Improves DenseNet Robustness**
>
> Thank you for your careful analysis of our paper.
>
> 1. In choosing our base augmentation operators we opted to reuse the operators in AutoAugment for the sake of simplicity, while taking out the ones which appeared in the ImageNet-C test set. As for the scalar hyperparameters, we have launched several ImageNet sweeps perturbing depth, width, count, and beta/dirichlet coefficients. We aim to share these results with you soon, and these experiments are so far indicating that, across different hyperparameter choices, AugMix is quite stable.
>
> 2 and 4.
> Thanks for pointing out the paper by Guo et al., 2019 [1]. We have cited it in the revision and added a discussion in the related work.
>
> We would like to clarify that our method does not mix images of multiple classes together, but rather the three augmentation chains are created from a single image and mixed back into one single image. We should like to note that the AugMix pseudocode and Figure 4 illustrate that the label remains constant through the augmentation process. Since the base transformations are label preserving, and we only mix different augmentations of the same image, we do not believe the manifold intrusion phenomenon presents a significant issue to our method unlike mixup, as most of the operations are fairly structure-preserving. One might expect a performance drop coincident with manifold intrusion, but AugMix increases performance on both clean and corrupted inputs. In view of your concern, we looked at several example images from AugMix and did not observe class collisions. However, we agree that the concept of manifold intrusion from Guo et al., 2019 for mixup is a real concern.
>
> 3. Additional discussion
>
> One possible explanation for the performance drop of “AugMix on top of Mixup” is that mixup is not a label preserving transformation, and applying augmix on top of mixup could suffer from manifold intrusion, thereby causing error rate to increase. We have added a comment linking to the relevant explanation from Guo et al., 2019 in the paper. Mixup alone substantially harms calibration as well, which means combining it with AugMix would make uncertainty estimates worse too. However, In Table 2 we show that combining AugMix with another label preserving augmentation such as SIN, outperforms both AugMix and SIN individually. Hence AugMix can combine with well with other techniques.
>
> Future work on extending AugMix by including ideas from n-fold Mixup (Guo et al. 2019) to avoid manifold intrusion could yield further benefits.
>
> 5. Additional experiments
> “Does the method work for other network architectures such as DenseNet?”
> To test robustness across network architectures, we report results with AllConvNet, Wide ResNet, and ResNeXt in Tables 1, Table 5, Table 6, and we observe that AugMix significantly improves performance across different architectures.
> On DenseNet, we observe that the error rate greatly decreases from 30.7% (baseline) to 12.7% (AugMix). We have added DenseNet results (Table 1, 5, 6) in the updated draft, thanks to your suggestion.
>
> We hope we were able to address your valid concerns and we thank you for your helpful suggestions. Do you have any remaining concerns?
>
> [1] Hongyu Guo, Yongyi Mao, Richong Zhang. MixUp as Locally Linear Out-of-Manifold Regularization. Proceedings of the AAAI Conference on Artificial Intelligence, 2019.

---

> > ### Comment · AnonReviewer1 · 2019-11-13
> > **thank you for your rebuttal and further questions**
> >
> > Thank you for your rebuttal. I really appreciate the additional experimental results and analysis, which I found very helpful. Below please find my further comments after reading your rebuttal and the other reviews.
> >
> > 1. Your observation of “Mixup alone substantially harms calibration” could be further explained or discussed. As shown in Thulasidasan’s NeurIPS19 paper "On Mixup Training: Improved Calibration and Predictive Uncertainty for Deep Neural Networks", Mixup does significantly improve the model calibration and predictive uncertainty.
> >
> > 2. The paper could be further improved if the authors could provide more insight or explanation on why the proposed method works. In your rebuttal, you did mention that “augmentations produced by AugMix are more "diverse"” may be one of the reasons for the improved performance of the proposed method. I wonder why "diverse" here is a good thing?

---

> > > ### Author Response · Authors · 2019-11-14
> > > **New Appendix B Added to Further Explain Why AugMix works**
> > >
> > > Thank you for your responsiveness and your reply. We have added Appendix B which hopes to offer some additional insight into the mechanisms behind AugMix’s performance.
> > >
> > > 1. “Calibration of mixup”:
> > > Sorry for the confusion, we realize now that our previous comment above “Mixup alone substantially harms calibration” may have sounded broader than we intended (note this comment was only mentioned in the response above, and not in the paper, so no changes are required in the paper).
> > > There are some differences between our setup and that of Thulasidasan et al. [1]. They are as follows.
> > > * Calibration on i.i.d. data vs calibration under shift: First, we consider calibration under unforeseen corruptions, while their analysis is performed on clean data alone. Ovadia et al. 2019 also show that calibration in i.i.d. setting does not always translate to calibration under shift.
> > > * Tuning: We also note that [1] tunes the alpha coefficient and in Figure 2h of [1] ( https://arxiv.org/pdf/1905.11001.pdf#page=5&zoom=100,0,89 ) we can see that the model becomes increasingly miscalibrated as alpha approaches the value recommended in [2]. An additional difference is that [1] uses a non-standard weight decay coefficient with Mixup. We use 1e-4, following [2] and [3], but [1] uses 5e-4 weight decay; we have found that 5e-4 weight decay noticeably increases error with Mixup, and [4] also notes that stronger weight decay can influence calibration. We have added in our paper that we use a standard weight decay coefficient as in [2, 3] per your advice.
> > >
> > > 2. “I wonder why "diverse" here is a good thing?”
> > > We note that previous works [5,6] have shown that if image modifications are not sufficiently diverse then the network will memorize and overfit to the specific and narrow distribution of modifications seen during training. To attain generalization, it is important to increase the variance of the augmentation distribution which we achieve through much stochasticity.
> > >
> > > “why the proposed method works.”: We have added a new Appendix B to give more analysis into how AugMix works. In addition to diversity of augmentations and the explanation in Appendix B, AugMix also enforces consistency between augmentations of the same image, which can be thought of as a way to encourage invariance in classifier predictions with respect to augmentations that preserve semantics. Our ablation experiments in Table 4 show the relative contributions of these two ingredients. We hope this explanation and the newly added Appendix B and Figure 9 shed light on how AugMix provides robustness.
> > >
> > > We hope we were able to address your valid concerns and we thank you for your helpful suggestions.  Do you have any remaining concerns?
> > >
> > > [1] On Mixup Training: Improved Calibration and Predictive Uncertainty for Deep Neural Networks. Sunil Thulasidasan, Gopinath Chennupati, Jeff Bilmes, Tanmoy Bhattacharya, Sarah Michalak. NeurIPS 2019.
> > >
> > > [2] Zhang, Hongyi, Moustapha Cisse, Yann N. Dauphin, and David Lopez-Paz. "mixup: Beyond empirical risk minimization." ICLR (2018).
> > >
> > > [3] Hongyu Guo, Yongyi Mao, Richong Zhang. MixUp as Locally Linear Out-of-Manifold Regularization. Proceedings of the AAAI Conference on Artificial Intelligence, 2019.
> > >
> > > [4] On Calibration of Modern Neural Networks. Chuan Guo, Geoff Pleiss, Yu Sun, Kilian Q. Weinberger. ICML 2017.
> > >
> > > [5] Examining the Impact of Blur on Recognition by Convolutional Networks. Igor Vasiljevic, Ayan Chakrabarti, Gregory Shakhnarovich.
> > >
> > > [6] Generalisation in humans and deep neural networks. Robert Geirhos et al. NeurIPS 2018.

---

> > > > ### Comment · AnonReviewer1 · 2019-11-14
> > > > **Thanks, but with further question**
> > > >
> > > > Thank you for your clarification. That helps.
> > > > Also, I found the newly added Appendex B was interesting and useful.
> > > >
> > > > Here is one more question if you don't mind.
> > > >
> > > > Since there is no guarantee (at least I did not see that) that the proposed method will work. I wonder when you would expect your approach to fail, namely degrading the performance of the baseline model? As an example, the Mixup could suffer from the manifold intrusion problem.

---

> > > > > ### Author Response · Authors · 2019-11-14
> > > > > **Reply**
> > > > >
> > > > > Thank you for your responsiveness and your reply.
> > > > >
> > > > > We agree that pointing out limitations is important. One potential limitation of most proposed data processing techniques for natural images, including ours, is that there is not an obvious extension of the entire method to natural language processing. However, at the very least our proposed Jensen-Shannon consistency loss could potentially be useful for NLP.
> > > > > Guo et al. [1] points out a potential issue with manifold intrusion. A possible problem with using data processing techniques including AugMix is that practitioners may include numerous additional augmentations, some of which could potentially change the class and intrude the manifold. We analyze an instance of manifold intrusion [1] in the newly updated Appendix C (Figure 11), as this is an important caveat to mention to the readers. However, we think AugMix in its current form does not have the issue of manifold intrusion since images of different labels are not mixed and the set of augmentations we chose does not change the class like in Figure 11. On all the problems we have tried so far (CIFAR-10, CIFAR-100, ImageNet), AugMix consistently improves performance.
> > > > >
> > > > > We thank you for your responsiveness. Do you have any remaining concerns?
> > > > >
> > > > > [1] Hongyu Guo, Yongyi Mao, Richong Zhang. MixUp as Locally Linear Out-of-Manifold Regularization. Proceedings of the AAAI Conference on Artificial Intelligence, 2019.

---

> > > > > ### Author Response · Authors · 2019-11-15
> > > > > **Double Check**
> > > > >
> > > > > Reviewer 1,
> > > > >
> > > > > Thank you for your responsiveness throughout this week and helping us improve our paper.
> > > > >
> > > > > After tomorrow OpenReview will not allow us to answer further questions or address other concerns.
> > > > > Do you have any further experiments you should like us to run for you or have any other questions?

---

> > > > > > ### Comment · AnonReviewer1 · 2019-11-15
> > > > > > **Thank you**
> > > > > >
> > > > > > Thank you for your rebuttal and the insightful comments to my questions. I really appreciate that.
> > > > > > I think the rebuttal has addressed some of my main concerns and I will adjust my score accordingly.
> > > > > >
> > > > > > Also,  I wonder if it would be a good idea to be a bit specific on "Jensen-Shannon consistency loss could potentially be useful for NLP" and include that in the conclusion/future work section of your paper?

---

### Author Response · Authors · 2019-11-11
**New Version of the Paper**

The paper has been updated and includes hyperparameter sensitivity analysis and DenseNet results. Hyperparameter results are in Appendix A, and the results indicate that, as hyperparameters vary, AugMix performance is relatively stable. The new DenseNet results show that AugMix reliably improves corruption robustness, perturbation stability, and uncertainty estimation.

---

### Decision · Program_Chairs · 2019-12-19

**Decision:**

Accept (Poster)

**Comment:**

This paper tackles the problem of learning under data shift, i.e. when the training and testing distributions are different. The authors propose an approach to improve robustness and uncertainty of image classifiers in this situation. The technique uses synthetic samples created by mixing multiple augmented images, in addition to a Jensen-Shannon Divergence consistency loss. Its evaluation is entirely based on experimental evidence.

The method is simple, easy to implement, and effective. Though this is a purely empirical paper, the experiments are extensive and convincing.

In the end, the reviewers didn't show any objections against this paper. I therefore recommend acceptance.